# Surface Design Options in Polymer- and Lipid-Based siRNA Nanoparticles Using Antibodies

**DOI:** 10.3390/ijms232213929

**Published:** 2022-11-11

**Authors:** Michael Gabel, Annkathrin Knauss, Dagmar Fischer, Markus F. Neurath, Benno Weigmann

**Affiliations:** 1Medical Clinic I, Research Campus, University Hospital of Erlangen, Hartmannstraße 14, 91052 Erlangen, Germany; 2Medical Immunology Campus Erlangen, Medical Clinic 1, Friedrich-Alexander University Erlangen-Nürnberg, 91054 Erlangen, Germany; 3Department of Chemistry and Pharmacy, Division of Pharmaceutical Technology and Biopharmacy, Friedrich-Alexander University Erlangen-Nürnberg, 91054 Erlangen, Germany; 4Medical Clinic I, University Hospital Erlangen, Ulmenweg 18, 91054 Erlangen, Germany; 5Deutsches Zentrum Immuntherapie (DZI), 91054 Erlangen, Germany

**Keywords:** siRNA, gene delivery, surface-functionalized nanoparticles, targeted drug delivery, lipid nanoparticle, polymer nanoparticle, antibody

## Abstract

The mechanism of RNA interference (RNAi) could represent a breakthrough in the therapy of all diseases that arise from a gene defect or require the inhibition of a specific gene expression. In particular, small interfering RNA (siRNA) offers an attractive opportunity to achieve a new milestone in the therapy of human diseases. The limitations of siRNA, such as poor stability, inefficient cell uptake, and undesired immune activation, as well as the inability to specifically reach the target tissue in the body, can be overcome by further developments in the field of nanoparticulate drug delivery. Therefore, types of surface modified siRNA nanoparticles are presented and illustrate how a more efficient and safer distribution of siRNA at the target site is possible by modifying the surface properties of nanoparticles with antibodies. However, the development of such efficient and safe delivery strategies is currently still a major challenge. In consideration of that, this review article aims to demonstrate the function and targeted delivery of siRNA nanoparticles, focusing on the surface modification via antibodies, various lipid- and polymer-components, and the therapeutic effects of these delivery systems.

## 1. Introduction

After the discovery of RNA interference (RNAi) in 1998 [1], two decades had to pass before the full potential of siRNA therapeutics was recognized and the first drug based on this technology, called Onpattro (patisiran, ALNTTR02), found therapeutic application in the clinic in 2018 [2].

The history of the discovery of RNAi already started in 1990, when Napoli et al. tried to increase the expression of the CHS gene in petunias to achieve a stronger flower pigmentation by adding additional copies of the CHS gene into the plants. Contrary to their predictions, the colour faded due to an alleviated pigment formation in the genetically modified plants. The phenomenon was named “co-suppression” because the additional genes also reduced the expression of the naturally occurring CHS gene [3]. Further research showed that the inhibition of the genes does not take place at transcription level and that increased formed mRNA degrades rapidly. This effect was called “Post-Transcriptional Gene Silencing” (PTGS) [4]. Finally, in 1998, it was demonstrated by Fire et al. in the nematode “*Caenorhabditis elegans*” that the produced mRNA itself is involved in PTGS and that efficient and specific gene knockdown is possibly caused by the delivery of double-stranded RNA [1]. Based on their pioneering findings, Andrew Fire and Craig Mello were awarded with the Nobel Prize in Physiology or Medicine in 2006 for the discovery of RNA interference [5]. In summary, starting with the silencing of genes in plants in 1990 [3] and in nematodes in 1998 [1], the technique evolved further, leading to the silencing of genes in different mammalian cell lines in 2001 [6]. Currently it is possible to block almost any gene of interest in humans by double-stranded RNA sequences or the significant smaller small-interfering RNA (siRNA), resulting in tremendous potential for RNAi usage [7,8,9].

Explained briefly, siRNA is able to abrogate the expression of selected genes through the mechanism of RNAi [10]. The potential in RNAi is impressive, since it is a highly guarded and conserved mechanism that can be found in eukaryotic cells, where it serves as a natural defender against foreign and potential dangerous nucleic acids entering the cell [7,11]. Furthermore, the process is so efficient that about 2000 siRNA molecules per cell are sufficient for an efficient response. Using nucleic acids for “silencing” genes comes with benefits. It is convenient for the development of new drugs, as it enables a re-targeting without big alterations of the drug formulation. Theoretically, the site of action and the pharmacological effect of the drug might be “customized”, switching off the gene of interest by replacing the siRNA sequence [11]. Furthermore, siRNA can easily and reproducibly be synthesized at large scale.

Multiple disadvantages and barriers in the handling and application of siRNA such as poor stability, rapid degradation by enzymes, low uptake into the desired target tissue, and insufficient encapsulation efficacy as well as effectiveness in transfection were overcome by new findings and advances in the application of siRNA through the use of appropriate surface functionalized drug delivery systems [9].

In order to precise the application of nanoparticles at the desired tissue, it makes sense to target molecular structures through the surface functionalization of nanoparticulate systems [12]. With further attention to development and research, the probability that the product will reach the daily clinical practice will strongly increase. In addition, the efficacy of nanocarrier-based drug delivery systems mainly depends on their controlled interactions with biomolecules. To enhance the efficacy of NPs and increase the delivery rate to targeted tissues, surface modification of particles is a promising option [8,12].

This review is divided into three sections to give a comprehensive overview of the three major aspects of specific siRNA application regarding pharmacology, delivery and targeting. Starting with the mode of action and barriers of siRNA, the second chapter deals with different lipid- and polymer-based delivery systems of siRNA. Finally, the last part of this paper illustrates the possibility of active targeting using antibodies and highlights their clinical effects by providing several examples.

## 2. Silencing of Genes: Fate of the siRNA

Gene therapy enables a versatile way for patients with life threatening diseases to get a treatment that better suits the individual and works in a more specific fashion. With the delivery of nucleic acids, the effects of genes may not only be turned off but can also be augmented or changed in the desired tissues. On the one hand, genes can be inhibited using small interfering RNA (siRNA), microRNA (miRNA) and inhibitory antisense oligonucleotides (ASOs) [13], which can silence the translation of the protein. On the other hand, DNA plasmids or messenger RNA (mRNA) can be introduced into the cell to promote or alter the expression of genes resulting in an increased synthesis of the targeted protein [14]. Many of the above-mentioned methods are applicable therapeutically, but we would like to take a deeper look at the system of RNA interference.

### 2.1. Challenges and Barriers for siRNA

Following intravenous application, siRNA is circulating in the bloodstream. In order to be effective, siRNA must leave the bloodstream and cross the cell membrane to enter the cytosol of the cell in an intact manner by overcoming the endosomal/lysosomal compartment [15]. Notably, this turns out to be difficult for the circulating siRNA due to its physical and chemical properties, since both the siRNA, as well as the cell membrane, are negatively charged, leading to a repulsion between the RNA and the membrane [7]. In addition, nonspecific bindings by plasma proteins further complicate the distribution of siRNA to its target site. Furthermore, the size of siRNA with a length less than 8 nm and a diameter smaller than 3 nm is still too large to penetrate the cell membrane directly [16]. However, the size is small enough to be excreted within a few minutes glomerularly by the kidney and to accumulate in the urine, which is another disadvantageous factor with regard to effectiveness [17,18]. In addition to the rapid clearance in the bloodstream, siRNA is exposed to RNAses in the plasma and tissue. Nucleases can cut the siRNA rapidly into inactive fragments with a higher efficiency than for DNA due to the 2′-hydroxy group of the RNA. siRNA is a target for defence mechanisms of the host, not only extracellularly but also intracellularly. Thus, siRNA taken up by endocytosis must first escape out of the endosomal and lysosomal vesicles to act in the cytosol [19]. In this context, siRNA packed into nanoparticles might be beneficial, enabling endosomal release via the proton sponge effect or preventing endosomal uptake of the siRNA in general via fusion or membrane penetration directly [19,20].

### 2.2. Mechanism of Action within the Cell

After the siRNA successfully managed the transport into the cell, the mechanism of RNAi as a natural defence mechanism against potential dangerous nucleic acids is activated, as shown in Figure 1 [21]. There is no difference whether a short sequence of siRNA or a longer double stranded RNA (dsRNA) is detected in the cytosol of the cell, as longer dsRNA is cut by the endoribonuclease dicer into small 20–25 base pair siRNA sequences [22,23].

Unlike a longer piece of dsRNA, the direct introduction of siRNA avoids a possible innate immune reaction with interferons caused by the interaction with intracellular receptors or by the activity of the enzyme dicer itself [24]. In the cytoplasm, double stranded siRNA binds to the RNA-induced silencing complex (RISC). A further component of the RISC complex called Argonaut 2 (AGO2) unwraps the double stranded siRNA in a single siRNA strand and removes the unnecessary “passenger” strand [25]. The activated RISC complex contains the antisense strand of the siRNA and guides the single-stranded antisense RNA inside the complex to the matching complementary mRNAs of the cell [26].

If base pairing between the antisense strand inside the activated RISC complex and the complementary mRNA strand matches, the RISC complex cleaves the mRNA and the fragments get degraded. As a result, the initial protein can no longer be translated [22,23,26].

After cleaving the target mRNA, the RISC complex with the included single-stranded siRNA can slice additional molecules of the target mRNA [27].

Thus, with only a few siRNA molecules in the cell cytosol, it is possible to slice an enlarged number of mRNA molecules within the cell, which represents the amplifying effect of the RISC complex [15], highlighting why the used siRNA has no stochiometric ratio to the degraded mRNA in the cell. The RISC mechanism further explains why it is possible to reliably silence genes by siRNA.

Bartlett et al. outlined that the silencing effect caused by siRNA lasts in rapidly dividing cells for several days and in slow dividing cells for up to several weeks. The activated RISC complex is stable for weeks; however, its concentration decreases with each cell division [28].

Showing that the amount of administered siRNA exerts a significant effect on the magnitude of gene silencing, but only a small effect on the duration of it, Bartlett et al. transfected non-dividing cells with four different siRNA concentrations (10 nM, 25 nM, 50 nM, 100 nM). They noticed significant differences in the magnitude of gene silencing, but only small alterations in the gene silencing duration. In addition, to show that the division rate of the cells has a significant impact on the gene knockdown duration, they used 4 different cell lines with different doubling times: Neuro2A-Luc (0.8 days), LNCaP-Luc (1.4 days), HeLa-Luc (1.6 days) and CCD-1074Sk-Luc (non-dividing). They demonstrated that cells with a high doubling rate had a shorter duration of gene silencing due to a dilution effect resulting from cell division [28].

### 2.3. siRNA in Function of a Drug: Effects and Side Effects

Using siRNA in a function like a drug has further advantages. Contrary to small molecular drugs or monoclonal antibodies, siRNA needs a full complementary Watson-crick base pairing, in other words a perfect nucleotide base pairing, to their matching opposite mRNA to exert the pharmacological effect. This makes the mechanism of siRNA very specific. In contrast, small molecular drugs must be able to recognize the complex spatial arrangement of their target, and moreover antibodies must be able to recognize the epitope of the active target structure in order to be effective. Unfortunately, not every disorder can be treated with small molecular drugs or monoclonal antibodies to date considering the lack of molecular targets [7].

At the same time, it has to be mentioned that with a few nucleotide mismatches at the complementary base pairing, the risk of off-target effects of siRNA increases, although single mismatched base pairings are expected to be tolerated by the RISC complex. In the case of larger contiguous mismatched base pairs, however, unspecific off-target effects can occur such as interferon immune responses by activation of Toll-like receptor 3 (TLR3) or TLR7 [29]. TLR3 activation can cause cell apoptosis, as Cho et al. showed in a neovascularization mouse model, in which non-targeted siRNA suppresses hemangiogenesis and lymphangiogenesis [30]. Furthermore, there may be a change in the expression of other undesired genes leading to non-specific effects [31]. Additionally, an unintentional knockdown of other genes can occur if the base pairing coincidentally matches the sequence of the siRNA incorporated in the RISC complex, resulting in degradation of the matched mRNA [32]. It seems that anywhere from 7 to 11 mismatched contiguous base pairings in siRNA can induce the translation repression of mRNA [32,33]. Nevertheless, siRNA is highly specific for effective gene silencing and has predominantly only one mRNA target, therefore, it is suitable for an effective suppression of mRNA translation.

Significant progress was made in the prevention from off-target effects and optimizing the safety of siRNA utilization. This includes the selection of sequences having only marginal overlap in the nucleotide matching with other mRNA sequences, or the siRNA structure can be modified chemically [34,35]. One example for an effective way to avoid off-target effects can be achieved by using siRNA bundles. SiRNA bundles are a whole series of different enzymatically generated siRNA sequences with the same target gene. The use of siRNA bundles shows a very potent knockdown without showing detectable off-target effects [36].

Despite the enormous therapeutic potential of siRNA and the growing interest in this technology, there are only a few siRNA-based drugs that are approved for clinical application [37,38,39] or under clinical investigation [40,41,42]. In conclusion, with all the difficult drug properties of siRNA, the administration of blank siRNA is insufficient. To improve the pharmacological potential of siRNA, several nanoparticulate systems have been evolved which will be discussed further in the following section.

## 3. Types of Nanoparticulate Drug Delivery Systems

Nanotechnologies and their possible application in medical fields such as therapy and diagnostics but also in electronics, material sciences and chemistry have recently attracted more and more interest from both scientists and non-scientists. For the pharmaceutical development of new therapies, nanoparticles with a size ≤100 nm are required because they offer different and diverse aspects in their structure and characteristics due to their large surface area [43,44,45]. As mentioned previously, siRNA has to overcome a lot of barriers and hazards in order to fulfil the pharmacological effect in the cell. Therefore, researchers developed various techniques and formulations to encapsulate the siRNA to counteract barriers and hazards. Kim et al. highlighted the current trends for siRNA drug delivery platforms [46]. They outlined that lipid-based nanoparticles followed by polymeric-based nanoparticles are superior to other drug delivery systems, such as metal- (gold and iron oxide) or silicon-based systems. Consequently, this review will focus on the possible surface design options in lipid- and polymer-based nanoparticles, as they are the leading drug delivery systems for siRNA.

The formulation and application strategies of many nanoscaled drug delivery systems were adapted from DNA to RNA. Mechanistic analogies can be found between the two nucleic acids in the process of complexation due to the negative charge, protection from DNAses or RNAses, as well as the release of the nucleic acids into the cell, since in both cases DNA and RNA are double-stranded nucleic acids with an anionic phosphodiester backbone [47]. However, differences already become obvious when looking at the size, structure and chemistry of DNA and siRNA molecules. The weight of plasmid DNA with up to several thousand base pairs is several times larger than the weight of siRNA with about 20–25 base pairs, and therefore shows significantly more electrostatic interactions with polycations, which is an advantage for the nanoparticle formation of DNA. Likewise, the DNA backbone shows a significant improvement in stability due to the stable deoxyribose component of DNA, in contrast to the sensitive 2′-hydroxy group of ribose on the RNA backbone [7].

Nevertheless, there are already quite a few ways to stabilize siRNA by chemical modifications, such as the introduction of 2′-fluorine or 2′-methoxy groups, thus making the application of siRNA more feasible [48]. Considering the site of action of the two nucleic acids, DNA must be transported into the nucleus, whereas siRNA already reached its target in the cytosol. Therefore, a suitable transport vehicle for siRNA has the focus on a reliable release from the endosome into the cytosol. Additional properties, such as buffer capacities at endosomal pH values or possible lytic properties that allow an enhanced nucleic acid release from the endosome, offer advantages [49].

However, due to the higher instability of RNA resulting from the 2′-hydroxy group, the lower chain length and the number of anionic charges, a less efficient electrostatic interaction as well as nanoparticle formation occurs. Therefore, a direct transfer from DNA to RNA is not possible, and an adaption of the formulation processes is necessary.

### 3.1. Lipid-Based Nanoparticles

Lipid-based drug delivery systems are the most widely used and marketed systems that include a whole range of carrier types. The most advanced liposomes are large amphiphilic vesicles with a lipid bilayer having both a polar aqueous core and an unpolar surrounding lipid bilayer. More complex lipid nanoparticles such as solid lipid nanoparticles, nanostructured lipid carriers and electrostatic cationic lipid- nucleic acid lipoplexes have a more complex structure and enhanced properties in stability compared to liposomes [50,51,52,53]. Originally, LNPs were developed as vehicles for DNA-based drugs. Over time, they have proven to be reliable delivery systems for siRNA, as they are also able to entrap siRNA and thus protect the siRNA from degradation. The lipid matrix of the majority of siRNA LNPs is composed of cationic or ionizable lipids, neutral helper lipids, cholesterol and shielding lipids [54].

Cationic lipids such as DOTAP [55], DOTMA [56] or DMAPAP [57] can electrostatically form stable complexes with the negative siRNA due to electrostatic interactions, thereby enabling the capturing of siRNA in the LNP. However, the net cationic lipid systems suffer from the so called “polycationic dilemma”, as a high number of cationic charges favour the formation of complexes, but vice versa suffer from cytotoxic effects due to unspecific interactions with negatively charged cell membranes and blood components [58]. In that regard, research about pH-sensitive ionizable lipids was intensified [59]. Ionizable lipids such as DLin-DMA, DODAP or DLin-MC3-DMA offer a key advantage during the production of LNPs, because their charge can be varied [55]. They can be positively or neutrally charged depending on the pH value of the medium and allow reliable electrostatic complexation with siRNA, resulting in efficient encapsulation during manufacturing. This ability to change the charge at different pH values reduces the toxicity of ionizable lipids [60]. On the other hand, at acidic conditions in the endosomal/lysosomal compartments, ionizable lipids can use the ability to alter their charge and interact with negatively charged cell structures to destabilize the endosomal membrane and thereby promote the release of siRNA into the cell cytosol [61].

The contribution of helper lipids such as DSPC and cholesterol seems to be crucial for the stability of LNPs. A typical helper lipid that is widely used in siRNA LNPs is DSPC [62]; however, the mechanism of action has yet to be fully understood. The helper lipid is normally used in low molar concentrations in the LNP formulation and is presumed to interact with the lipid layer, thus stabilizing the LNP [63]. Another component that has a significant effect on lipid packing, membrane fluidity and permeability of the lipid bilayer is cholesterol [64]. By inserting cholesterol in the lipid formulation, it is possible to reduce the distance between the individual lipids in the lipid layer and thereby prevent the premature release of the active ingredient [65].

In addition, the characteristics of shielding lipids are interesting as well. Shielding lipids are often associated with the biocompatible hydrophilic macromolecular polymer poly (ethylene glycol) (PEG), which was initially invented to hide and protect biological substances due to the fact that they can be removed rapidly from the bloodstream by detection from macrophages [66]. PEG alternatives such as synthetic polymers (e.g., poly (glycerine), poly (acrylamide)), natural polymers (e.g., polyamino acids, glycosaminoglycans) or zwitterionic polymers (e.g., poly (carboxybetaine), poly (sulfobetaine)) are currently under intensive investigation due to PEG side effects such as the ABC phenomenon [67], the formation of PEG antibodies, as well as a low drug efficacy as a consequence of anti-PEG antibodies [68,69]. Further shielding of lipids increases the circulation half-life and stability and prevents the aggregation of particles during storage [70]. However, when using PEG, the so-called “PEG dilemma” is also encountered, whereby the efficient transfection of cells may be compromised because PEG shielding may not only protect the LNPs from the recognition by the immune system but also from the interaction with target cells [71]. As a consequence, the reduced transfection efficiency by PEGylation can be improved using special PEG lipids, where the linkage between PEG and lipid opens after administration time-dependent or in a stimuli-responsive manner and exposes the LNP surface, making them available for cell transfection once again [72]. Nevertheless, since the most common administration of LNPs is an intravenous injection into the body, resulting in a rapid systemic distribution, LNPs suffer from a short half-life when administered parenterally. To ensure a controlled or long half-life, the assistance of further delivery systems, such as hydrogels, for example, is recommended [73]. This allows the use of additional routes of administration to protect the LNPs from rapid clearance and to release it in a controlled manner from a drug depot at the target tissue [74]. Concluding LNPs achieve a high loading capacity for siRNA as well as an efficient transfection of cells; however, in the case of prolonged or delayed release, they are limited.

### 3.2. Polymer-Based Nanoparticles

Polymer NPs are the other main class for siRNA delivery. They represent a wide and diverse fraction of matrices with variations of each formulation regarding the type of encapsulation (nanospheres and nanocapsules). Polymers offer a reliable and proven way to control the release of drugs through their method of application (oral, parenteral, subcutaneous, local or systemic) and can be modified for sustained and controlled release [75,76]. Through the selection of various polymers in the carrier matrix, several different release mechanisms can be applied (diffusion- and solvent -controlled release, polymer-degraded or pH-sensitive release) [77]. This allows to influence the drug release and to adjust a therapeutic range and can prevent multiple drug administrations to the patient. At the same time, polymer NP offers high physicochemical stability, various targeting moieties through their diverse modification capabilities, as well as good transfection efficiency [78]. By using polymers, many formulations can be established, such as poly (lactid-co-glycolid) (PLGA) [79], poly (L-lysine) (PLL) [80], poly (L-arginine) (PLA) [81], poly (methacrylamide) [82], chitosan [83], poly (ethylene imine) (PEI) [84], cyclodextrin [85], hyaluronic acid [86], gelatine, [87] and alginate [88], which represent a selection of different polymers used in drug delivery systems.

These polymers can be divided by natural and synthetic origin. Thus, in the group of natural polymers, candidates can be found, such as polysaccharides [89], peptides [90], or bacterial nanocellulose [91]. The group of synthetic polymers includes representatives such as PLGA or poly (lactic acid) (PLA), amongst others [92]. Both types have their pros and cons. While natural polymers stand out due to their good biocompatibility and material availability, synthetic polymers often impress with their enormous variations and the possibility to produce them precisely and in a purified manner. The high purity of natural polymers is more challenging to achieve, as natural by-products may be present. The major disadvantage of synthetic polymers is their often problematic biocompatibility and biodegradability [93].

PLGA, for example, as a widely used polymer in drug delivery is already marketed in several applications, and is often adapted as a particle matrix. Due to its good biocompatibility and biodegradability, and potential to control drug release, it is applied in formulations approved by regulatory authorities [94,95]. However, the encapsulation of the water-soluble siRNA in the lipophilic PLGA is challenging. During production, siRNA can escape due to its hydrophilic properties and electrostatic repulsions, which are caused by the phosphate groups in the siRNA and the carboxylic residues of the PLGA polymer. As a result, only a small amount of siRNA can be entrapped inside the PLGA particles [95]. The use of cationic polymers such as PEI or oligopeptides (such as poly (beta-amino ester) to complex the nucleic acids into the polymer matrix can increase the entrapment efficiency [96,97].

Another innovative approach is to combine the advantages of both polymers and lipids, as shown by Ewe et al. [98]. By combining PEI/siRNA polyplexes with different phospholipid liposomes (consisting of the lipids DPPC, DPPC/DPPE or DPPC/DPPG), so called “lipopolyplexes” can be generated. These enable PEI to efficiently condense nucleic acids and at the same time to mask the cytotoxicity of the PEI polymer through good biocompatible properties of the lipids. In an ex vivo PC-3 prostate carcinoma xenograft model, both PEI/siRNA polyplexes and PEI/siRNA/DPPC lipopolyplexes were able to silence 50% of the survivin gene [98]. To summarize, in this study the benefit of polymers and lipids in combination provided a high knockdown efficiency with good biocompatibility.

Despite the challenges, advanced polymer formulations can be designed to achieve superior gene silencing effects, high cellular uptake, and a good endosomal escape rate, without any toxic incidents or problems in biosafety [99,100].

## 4. Surface Design Options

Most nanoparticles have to act peripherally from their application site, and specific targeting of the site of action by the drug delivery systems is crucial for optimal treatment success to avoid the undesired side effects in healthy tissues, to reduce drug dosage and to improve patient compliance.

Passive targeting takes advantage of the so called enhanced permeation and retention effect (EPR), a phenomenon whereby many solid tumors have a leaky vascular blood vessel as well as an incomplete lymphatic system, and nanoparticles tend to accumulate favorably in these incomplete vascular systems [101]. The effect can be optimized by surface modification of the NPs with hydrophilic macromolecules to achieve a prolonged circulation in the bloodstream [102]. Covalent adsorptive attachment of PEG to the surface of nanoparticles results in in vivo circulation and favored deposition in tumorous and inflamed tissues [103]. At the same time, PEGylation increases the stability in biological fluids and reduces the aggregation of particles during storage and the in vivo application [104]. Aldayel et al. developed a formulation for lipid-based siRNA NPs, taking advantage of the beneficial accumulation and targeting properties of PEG in inflamed tissues [105]. They functionalized the surface of the nanoparticles with an acid-sensitive PEGlipid called acid-sensitive stearic acid-polyethylene glycol (2000) hydrazone conjugate (PHC). Using this special stealth lipid, they targeted the lower pH microenvironment of chronic inflammation sites. The intravenous injection of acid-sensitive TNF-α siRNA lipid nanoparticles in arthritis mouse models resulted in significant reductions in paw thickness, bone loss, and histopathologic scores in an arthritis mouse model unresponsive to methotrexate therapy compared to pH- unsensitive nanoparticles. The acid-sensitive PEG lipid TNF-α siRNA formulation illustrates the ability to respond to a lower pH microenvironment and target chronic inflammation sites as well as showing a therapy option when unresponsive to methotrexate therapy [105].

### 4.1. Active Targeting for siRNA-Loaded NP

The ability to design nanoparticles that specifically recognize and target structures sounds ambitious. A number of different nanoparticle types with unique physicochemical attributes or programmed behaviours, such as nanoparticles that respond to light [106], ultrasound [107] or heat [108] to disrupt cells or release the content of the NP have been designed. Furthermore, nanoparticles can be modified to carry multiple drugs simultaneously [109], to release the contents upon some trigger (for example changes in pH) [110], or to contain a combination of diagnostic and therapeutic agents called theranostics [111]. An alternative way to specifically transport the drug to the site of interest involves the functionalization of the NP surface using different ligands. In consequence, target structures in tissues or on cells can be precisely addressed by interacting with cell-specific ligands on the surface of the NP. In most cases, biological ligands are used, such as aptamers [112], peptides [113], polysaccharides [114], small molecules [115], antibodies [116], receptors [117] or antibody fragments [118]. Despite the promising features of active targeting, the effect should not be overestimated or misunderstood, as Wilhelm at al. reported after studying the literature on nanoparticle delivery in mouse models over the last 10 years. They concluded that only 0.7% of the injected nanoparticle dose managed to find a way into the tumor tissue [119]. Nanoparticles cannot be directly steered in the sense of a guided missile, as Bartlett et al. reported, but active targeting is able to outperform untargeted strategies [120]. By functionalizing the surface of the nanoparticles with transferrin (Tf), the Bartlett was able to show a 50% increase in tumor luciferase activity in mice treated with Tf-targeted NP compared with untargeted NP treated mice, suggesting that a higher amount of siRNA reaches the tumor tissue and that the Tf-targeted formulation is more effective compared with non-functionalized nanoparticles. This leads to the conclusion that surface functionalization can have a crucial influence on the binding and uptake of the particles as soon as they reach the target tissue [120]. Thus, NPs functionalized with ligands influence the amount of nanoparticles localized within cancer cells, as illustrated by electron micrographs from tumor tissue suggesting an improved therapeutic effect [121]. In conclusion, there are still major hurdles to overcome in active targeting in order to significantly outperform existing methods. Provided that only <1% of the administered nanoparticles reach the target tissue, this implies that with higher delivery efficiency, the current high costs of nanoparticle production can be lowered as the administered NP dose sinks. Furthermore, it is evident that the remaining particles that do not reach the target tissue (~99%) will settle in peripheric tissues, thereby presenting a risk for adverse effects. Finally, this marginally low delivery rate underlines that the understanding of the exact process of delivery is still uncertain, and further knowledge is needed to get the delivery rate significantly above ~1% [119].

### 4.2. Nanoparticle Functionalization with Antibodies

Antibodies, also known as immunoglobulins, are glycoproteins produced by our body and serve as a tool for our immune system to detect and tag any exogenous antigen. Thus, they developed quickly as biologicals addressing specific structures in the body [122]. Using antibodies as ligands to address specific structures has also been established and finds application in the field of therapeutic nanoparticles (see Figure 2), as described in our review.

Antibodies can basically be divided into a number of different immunoglobulins (IgG, IgA, IgM, IgD and IgE) which all have different properties. Normally the group of IgG immunoglobulins finds therapeutic use in the functionalization of surfaces. The structure of IgG antibodies can be characterized as a Y-shaped structure and can be further subdivided into heavy and light chains. In turn they can be classified into further subdomains which are bound together by various functional groups [123]. A differentiation between the antigen-binding fragment (Fab) and fragment crystallizable (Fc) is possible. Both parts of the Fab fragment form the antigen binding site and enable specific antigen recognition [124]. The Fc region provides information about the origin of the antibody and can react with specific Fc receptors on the surface of immune cells, thus activating the immune system. By cleavage of the antibody into its components, antigen-binding fragments (fab) and single-chain variable fragments (scFv) with very specific pharmacokinetic properties can be generated with high specificity and a lack of Fc immunogenicity [125]. To increase the targeting capabilities of nanoparticles, they can be functionalized with antibodies or antibody fragments [126]. The orientation of the antibodies is crucial in this process. The successful immobilization of antibodies on a surface increases the biological activity of the antigen recognition of nanoparticles. The functionalization of the antibody can be randomly or specifically oriented. Optimal orientation occurs when the Fab region is not involved in the functionalization and is free for antigen recognition [127]. This is the case when the Fab region is freely available and not sterically hindered. Another factor influencing successful surface functionalization is the density of antibody molecules on the surface [128]. An excessive density can lead to a more difficult accessibility of the antigen recognition site, and the effectiveness of the antibodies decreases. In addition, the type of binding of the antibody to the nanoparticle surface also influences its effect. Only through stable binding can the targeting effect of the antibody be guaranteed. The possibilities of binding an antibody on the surface represent physical or ionic adsorption, a covalent bond, or the use of compound molecules [129]. Each of the options has its own advantages, disadvantages as well as an impact on the orientation of the antibodies on the surface. In order not to go beyond the scope of this review, we refer to the following paper from Marques et al. [130] if there is an interest in the different ways of immobilizing antibodies and antibody fragments on the surface of NPs.

### 4.3. Examples for the Use of Surface Functionalised siRNA NPs and Their Therapeutic Effects in the Treatment of Cancer and IBD

In 2020, approximately 19.3 million people were diagnosed with cancer, and about 10 million people died from it, while at the same time about seven million patients worldwide (status 2017) suffer from IBD, and it is assumed that for both diseases that the number of cases will increase in the coming years, which underlines the need for prevention of cancer and IBD, but also the development of new therapeutic options [131,132]. The power of RNAi using appropriate drug delivery systems may be the key to new drugs for these diseases, as the development of new RNAi strategies has been driven by the increasing interest in the ability to precisely control gene expression. By modulation of damaged genes or enhancing the formation of therapeutically relevant genes, the human genome can be therapeutically influenced [133] to prevent the aforementioned diseases. Interesting approaches to prevent or treat cancer and IBD using surface-modified siRNA nanoparticles will be presented in the next sections, distinguishing between lipid- and polymer-based approaches.

### 4.4. Polymer-Based siRNA NPs

Referring to polymer-based approaches, a representative polymer that can bind to the cell surface receptor CD44 is hyaluronic acid (HA) [134]. CD44 expression correlates with many subtypes of tumors and it can thus be a marker for cancer stem cells [135]. At the same time, HA can be used for NP production [136] and for surface modification with the aim of targeting cancer stem cells.

Choi et al. showed how to use HA as a compound of the layer-by-layer NP formulation and to simultaneously use its active targeting effect against CD44 [136].

They covalently conjugated HA, which forms the outer layer of their polymer-based NP, consisting of PLGA, poly (L-arginine) and HA with anti-CD20 antibodies to increase the active targeting effect in a binary way. Moreover, the binary targeting of Toledo (human non-Hodgkin’s B-cell lymphoma) cells with their nanoparticles produced great results, whereas the transfection with conventional transfection agents (Lipofectamine RNAiMAX and HiPerFect) was unsuccessful in this challenging cell line. With the used siRNA inside of their CD20/CD44 targeted particles against the pro-survival protein B-cell lymphoma 2 (BCL-2), they were able to impair the proliferation of firefly luciferase expressing Toledo cells (nine-fold decrease in luminescence intensity), which were injected into the tail vein of SCID beige mice for preparing the orthotopic non-Hodgkin’s lymphoma (NHL) animal model. The way of targeting two cell structures (CD20/CD44) concurrently on a cell type that is known to evade transfection is advantageous. The dual anti-CD20/CD44 formulation provided a survival benefit in the orthotopic NHL model. 100% of the dual-targeted CD20/44-BCL-2 siRNA loaded NP treated mice group survived the whole in vivo experiment time of 46 days, whereas the control group without NP treatment were euthanized after 23 days due to their severe health condition [136].

In summary, the binary targeting of blood cancer cells with double surface functionalized siRNA NP is very effective in the treatment of blood cancer and may be the basis for new treatment findings. Finally, the layer-by-layer formulation of this NPs using electrostatic interactions between different layers is also very interesting and worthy of mention [136].

Another different siRNA nanoparticle approach with the aim to address cancer cells through active targeting is the formulation from Guo et al. [137]. They address leukaemia stem cells (LCSs) with their polymer based-siRNA nanoparticles to treat acute myeloid leukaemia (AML). In this study they developed cyclodextrin- based nanoparticles (CD.DSPE-PEG-FAB) with a fragment antigen-binding (Fab) structure on the NP-surface that specifically targets the IL-3 receptor α-chain (CD123) of the human AML LSCs. The targeting of this antigen promoted an increased cellular uptake in KG1 cells in vitro (an AML cell line) and in samples from AML patients, and thereby enabled an efficient delivery of bromodomain-containing protein 4 (BRD4) siRNA. However, the silencing of BRD4 mRNA and protein levels in LSCs led to further myeloid differentiation, induced leukaemia apoptosis, and prevented the fulminant proliferation of cancer cells.

For generating the surface functionalized NP, Guo et al. first complexed the siRNA with an amphiphilic cationic β-cyclodextrin polymer (called “SC12-click-propylamine-CD”) [137]. Afterwards, DSPE-PEG-Fab was prepared by transferring the fab part from an IL-3Rα monoclonal antibody to the PEGylated linker (DSPE-PEG-maleimide) to obtain the finished component. Thereafter, a self-assembling process was used in which the additional component DSPE-PEG-Fab was added by “post insertion” to the CD-siRNA complexes.

Using fab modified NPs directed against IL-3Rα at a siRNA dose of 200 nM (siBRD4) in mononuclear cells from an AML patient resulted in a 40% reduction of BRD4 mRNA levels-and a 45% reduction of protein levels compared to the control siRNA formulation. There is also a significant difference in the knockdown effect compared to the conventional transfection reagent Lipofectamine 2000, which underlines the advantage of the surface modification with the fab fragment against IL-3Rα in this study.

In addition, Zhang et al. investigated a suitable formulation for cancer treatment [138]. The polymer used here consists of the compound carboxymethyl chitosan modified with histidine, cholesterol and anti-EGFR antibodies, abbreviated as “CHCE”. The surface functionalized NP complexed with siRNA against vascular endothelial growth factor A (VEGFA) caused cell apoptosis and inhibited proliferation of the xenograft tumor in an in vivo mouse model, induced by the application of the malignant melanoma cell line “SK-MEL-28”. In this study, the results illustrate that surface modification offers significant advantages in therapeutic applications [138].

Mice-bearing tumors treated with EGFR-targeted VEGFA siRNA NPs showed comparable body weight with the PBS-treated group, while having the lowest tumor mass compared to the group treated with untargeted VEGFA siRNA NPs. Regarding the release of inflammatory cytokines such as IL-6, IFN-γ or TNF-α, functionalization of the NP surface with EGFR-antibodies showed no significant difference compared to the PBS and untargeted NP group, so functionalization of the surface did not increase inflammatory cytokines here [138].

A further polymer-based approach of surface-modified siRNA NP is described by Xiao et al. [139]. However, in contrast to the previously mentioned ones, this study tried to target the colon of mice with IBD. Furthermore, this formulation is intended to be administered orally, whereas the others required i.v. administration. Here chitosan was modified with urocanic acid and PEG. In addition, single-chain CD98 antibodies (scCD98) were linked to the polymer using click chemistry for the active targeting strategy [139].

The resulting polymer (scCD98-PEG-UAC) was condensed with PEI and siRNA against CD98 (siCD98). The formed NPs (scCD98-functionalised siCD98 NPs) were embedded in a hydrogel of chitosan and alginate to allow oral administration [139].

Inhibition of CD98 on the surface of colonic epithelial cells and macrophages alleviates the severity of IBD. Experimental colitis models were chosen for functional in vivo experiments and demonstrated that the severity of colitis could be successfully treated with targeted siRNA NP administration. In an experimental T cell transfer model, the study group treated with scCD98-functionalized siCD98 NPs (treatment group) showed a significant reduction in weight loss compared to untreated mice (control group) or mice treated with scCD98-functionalized control siRNA NPs (treatment control group) [139].

Furthermore, the treatment group showed ~65.7% reduction in myeloperoxidase (MPO) levels, 65% reduction in CD98 mRNA expression, and a significant reduction in mRNA expression of inflammatory cytokines such as TNF-α, IL-6, and IL-12 compared to the untreated group. The results in the DSS mouse model showed comparable results. MPO levels decreased and mRNA expression levels of CD98 were reduced in the treatment group (47.7%) compared to the control group. Likewise, a reduction of inflammatory cytokines such as TNF-a, IL-6, and IL-12 was shown in this model [139]. This paper shows that the application of targeted siRNA NP via oral administration in a mouse model is successful.

### 4.5. Lipid-Based siRNA NPs

In order to treat human papillomavirus (HPV), which induces head and neck cancer (HNC), Kampel et al. designed a lipid-based siRNA NP formulation [140]. Here, siRNA is used against the viral proteins E6 and E7, as these two proteins play a crucial role in the replication of HPV viruses and the development of HNC. A knockdown of E6 and E7 in HPV-positive cervical and oropharyngeal cancer lines (UPCI:SCC090, UM-SCC-47, UD-SCC-2) can induce apoptosis [141]. The therapeutic effect in HNC treatment of patients using antibodies against the epidermal growth factor receptor (EGFR) has already been investigated [142]. Thus, this study attempted to combine both treatment approaches by concurrently using siRNA against viral proteins of HPV and the treatment with anti-EGFR antibodies functionalized on the NPs. Lipid nanoparticles containing siRNA against HPV-E6/E7 were prepared from the following lipids: DSPC, DMG-PEG, DSPE-PEG-Ome and the cationic lipid 10 EA-PIP, and the surface was modified by an antibody against EGFR. For the surface modification, a special targeting platform called “Anchored secondary scFv enabling Targeting” (ASSET) was used [143]. This involves the usage of a lipoprotein that self-inserts into the LNP surface and interacts with the fc domain of the antibody, enabling the conjugation of the antibody on the surface.

Moreover, a xenograft HPV-positive HNC mouse model was established and treated with the targeted siE6/siE7 LNPs. Surface functionalization with anti-EGFR antibodies improved the uptake of LNPs, especially into the cancer cell cytosol, and exerted a therapeutic anti-tumor effect by blocking the EGFR moiety in their mouse model. The surface functionalization of LNPs with anti-EGFR antibodies enhanced the therapeutic effect of apoptosis in vitro and led to a significantly better inhibition of tumor growth in the cancer mouse model compared to the LNPs modified by an isotype control antibody (isoLNPs). Treatment with targeted siE6/7 LNPS resulted in a 50% reduction of tumor volume compared to untargeted LNPs with control siRNA.

Kampel and her group demonstrated that the targeted LNPs are increasingly found intracellularly within the tumor cells, whereas the isoLNPs are predominantly found in the tumor stroma. In this case, it can be concluded that the surface modification with an antibody led to a stronger intracellular drug delivery in vivo, and therefore supports the synergistic therapeutic effect [140].

The group of Lu et al. also demonstrated the positive effect of surface modification [144]. They produced DOTAP/cholesterol liposomes using the lipid-film hydration method. TSPAN1 siRNA was loaded into the liposomes together with calf thymus DNA and protamine. DSPE-PEG-Mal was added to the prepared TSPAN1 siRNA loaded liposomes to bring active vinyl groups to the surface of the liposomes. These vinyl groups were able to react with the thiol groups of the TH17 antibodies and finally form Th17 targeted TSPAN1 loaded liposomes [144]. TH17 cells and the release of IL-17 play a crucial role in cancer progression [145]. The gene Tetraspanin1 (TSPAN1) is highly expressed in cancer and immune cells and is involved in immunoregulation and therefore represents a target gene which can be knocked out by siRNA [146]. In in vitro experiments, the Th17 targeted TSPAN1 siRNA LNPs were able to achieve an 80% TSPAN1 mRNA knockdown after 48 h with 200 nM siRNA in TH17 cells. TSPAN1 protein expression was also reduced by 80% in the targeted siRNA LNPs, whereas the siRNA LNPs without surface modification achieved only a 40% TSPAN1 protein reduction [144]. In the in vivo experiment using mice that spontaneously develop gastric adenocarcinoma, targeted siRNA LNP enabled a longer tumor-free time compared to the other groups.

In this study, a benefit from surface modification was also demonstrated in both in vitro and in vivo experiments using surface-modified siRNA liposomes.

Another example for the treatment of metastatic lung tumors using targeted siRNA LNPs is described in the paper of Lee et al. [147]. Initially, liposomes were made of the components O,O’-dimyristyl-N-lysyl glutamate, cholesterol, DSPE-PEG2000, DSPE-PEG2000-mal and Rho-DOPE. Cetuximab, an antibody against EGFR, was conjugated to the PEG-lipid exposed on the surface of the liposomes. Finally, the used anti-tumor siRNAs (Bcl-2 and survivin) were condensed with the cationic peptide 9-arginine and mixed with the anti-EGFR targeted liposomes to form the final anti-EGFR-9Arg-siRNA loaded lipoplexes. For generating mice with metastatic lung tumors, “LS174T-Luc” cells were directly injected into the left lateral thorax of mice and resulted in solid tumors 14 days after injection. Afterwards, mice were randomly divided into a control group and two treatment groups (one receiving the anti-EGFR targeted- and one receiving the untargeted siRNA lipoplexes), and received a total siRNA dosage of 1.5 mg per kg bodyweight three times weekly for 3 weeks. The advantage of surface modification by an antibody could be demonstrated in several ways. In mice with lung tumors, more targeted lipoplexes were localized in the tumor tissue compared to the group treated with untargeted LNPs. Furthermore, mice treated with targeted lipoplexes showed significantly weaker tumor signals in IVIS images. Finally, on day 23, all animals treated with the targeted lipoplexes were still alive, while the survival rate in the control group was 0% and in the untargeted NP group it was 33%. In addition, computer tomography scans confirmed the results from the IVIS by showing significantly slower tumor growth in the targeted lipoplex group than the other two groups. In conclusion, a significant inhibition of BCL-2 and survivin mRNA and proteins in the targeted LNP group was also seen in the tumor samples. In this study, it was shown that surface modification of siRNA-loaded lipoplexes by antibodies achieved a significantly better therapeutic effect and extended the lifespan of mice bearing lung tumors.

Veiga et al. demonstrated in their paper that the use of targeted LNPs can serve as a new immunomodulatory modality for the treatment of IBD and other inflammatory disorders [148]. They used DSPC, cholesterol, DMG-PEG200, DSPE-PEG200 and MC3-DMA to generate LNPs. They used the ASSET platform to functionalize the surface of the LNPs with antibodies against Ly6C to enable active targeting and selective manipulation of Ly6C+ inflammatory leukocytes and to reduce an unwanted knockdown in other cell lines. Concurrently, siRNA against interferon regulatory factor 8 (IRF8) was selected because the IRF8 protein plays a crucial role in the differentiation, polarization, and activation of leukocytes, making the inhibition of IRF8 a potential therapeutic target in the treatment of IBD. Finally, they established an experimental DSS mouse colitis model and injected the LNPs intravenously to demonstrate the therapeutic effect of the anti-LyC6 targeted IRF8 siRNA loaded particles (T-IRF8 LNPs) in vivo.

A significant reduction of TNFα (to baseline), IL6 (~60%), IL12 (~40%) and IL1β pro-inflammatory cytokines was detected in the T-IRF8 treatment compared to T-control siRNA treatment. The other groups received untargeted IRF8 siRNA LNPs, DSS or were left untreated. Furthermore, a significant improvement could be shown by colon length measurements (~20% improvement in colon length) and endoscopy (~35% decrease in MEICS score [149]) in the T-IRF8 group compared with the other groups [148]. This paper demonstrates good therapeutic effects through the intravenous administration of targeted siRNA loaded LNPs in peripheral local treatment of IBD in mice.

## 5. Conclusions

As this review has shown, there are already several surface functionalized siRNA NP formulations for specific targeting of different tissues in the literature (see Table 1). The formulations and strategies are quite variable, as shown by the selection of the polymer- or lipid-based approach and in the selection of the surface modification by the choice of the appropriate antibody (full antibody, antibody fragment or single chain antibody). Additional surface functionalization of NPs through antibodies is able to increase the therapeutic effect in experiments. The tissues addressed in the studies presented are predominantly cancer or immune cells that can be targeted in different ways. The surface functionalization of NPs makes an enormous contribution in optimizing the therapeutic effect. However, for a breakthrough in surface functionalized siRNA NPs, some parameters are very crucial, such as the choice of the best major target gene to be inhibited by siRNA and the selection of an advanced and safe delivery system, which protects the siRNA properly and at the same time releases it reliably at the target site. Moreover, a suitable ligand for the surface modification is crucial to avoid accumulation or binding of the NPs to undesired cell structures and thereby minimizing off-target effects. The spatial and time-controlled delivery of siRNA is the key for developing breakthrough RNAi-based medicine.

## 6. Outlook

siRNA in a function of a drug combined with advanced antibody functionalized nanoparticles demonstrates an innovative, safe, and robust strategy to silence gene expression in vivo, as the different studies and formulations in this review showed. This technique can serve as a valuable research tool for a wide range of applications and at the same time offers the possibility to create a breakthrough in the treatment of many diseases. With further progress and research in the field of surface functionalization using antibodies or antibody fragments, we are sure that there will be great novel findings in the upcoming years. Furthermore, we are confident that by combining new trends, such as theranostics, PEG alternatives or stimuli-responsive systems with RNAi in NPs simultaneously, both safety and specific targeting can be significantly enhanced by taking advantage of the benefits of different systems. Regardless, the way forward is challenging, and further research needs to be undertaken.

## Figures and Tables

**Figure 1 ijms-23-13929-f001:**
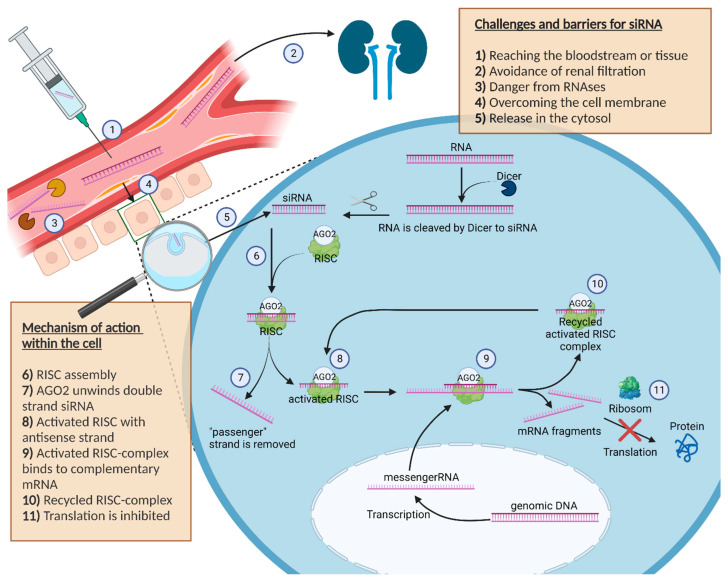
Schematic illustration about the challenges, barriers and the mechanism of action of siRNA in the cell. Furthermore, the figure shows the journey of siRNA starting at the injection site to the cell cytosol, where siRNA can exert its pharmacological effect. The description is in the box inside the picture.

**Figure 2 ijms-23-13929-f002:**
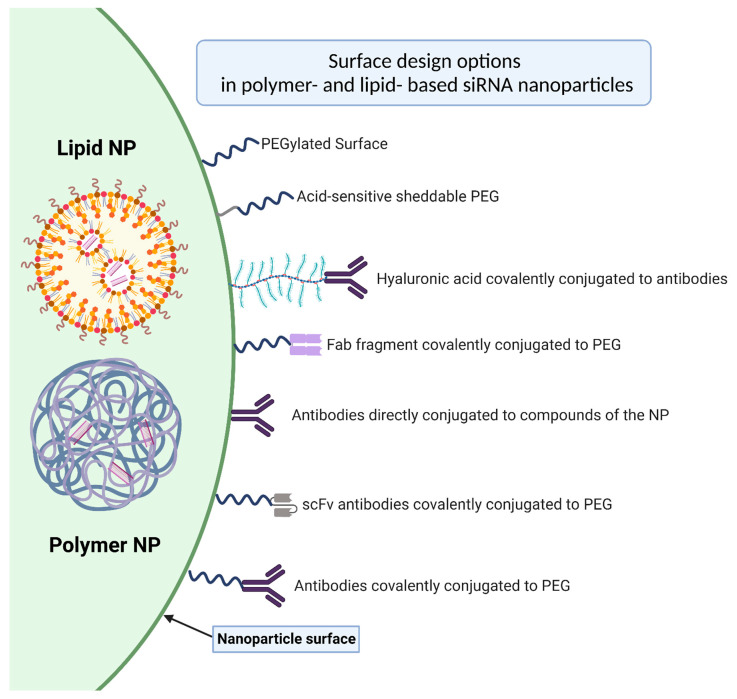
The schematic illustration shows possible surface design options in polymer- and lipid- based siRNA nanoparticles. For simplicity, no distinction was made between lipid- and polymer-NP. In addition, this illustration shows an optimal orientation of the antibodies and antibody fragments with an unhindered antigen recognition site. Furthermore, possible ways to conjugate antibodies or antibody components directly on the surface or on linking compounds of the surface are presented.

**Table 1 ijms-23-13929-t001:** Overview of several antibody functionalized polymer- and lipid-based siRNA NPs in the literature and their therapeutic effects.

Type of Nanoparticle	Main Components of the Formulation	Surface- Functionalization	Applied siRNA	Targeted Disease	Therapeutic Effect	Ref.
**Polymer-based Particle**	PLGA NP core covered by poly-L-Arginine-, siRNA- and hyaluronic acid layers	Hyaluronic acid (CD44-ligand) covalently conjugated to anti- CD20 antibodies	B-cell lymphoma 2 (BCL-2) siRNA	Hematologic Cancer Cells	Treatment with NPs induced apoptosis and impaired the proliferation of blood cancer cells in vitro and in an orthotopic non-Hodgkin’s lymphoma animal model	[136]
	Amphiphilic cationic β-cyclodextrin called “SC12-click-propylamine-CD”, Fab fragment bound to a PEGylated linker to get the DSPE-PEG-Fab component	Fab specifically targets the IL-3 receptor α-chain (IL-3Rα, known as CD123)	Bromodomain-containing protein 4 (BRD4) siRNA	Acute myeloid leukemia (AML)	Downregulation of BRD4 mRNA and protein in an AML cancer cell line in vitro and in AML patient derived samples ex vivo. This resulted in increased apoptosis and the impaired proliferation of leukaemia cells	[137]
	Carboxymethyl chitosan modified with cholesterol, histidine and antibodies	Anti-epidermal growth factor receptor (EGFR) antibodies	Vascular endothelial growth factor A(VEGFA) siRNA	Tumor treatment	Treatment with NPs induced cancer cell apoptosis and inhibited the proliferation of xenograft tumors in an in vivo mouse model	[138]
	Polymer formulation with proton-buffering groups based on chitosan and polyethylenimine. Embedded in a chitosan/ alginate hydrogel	Single-chain anti-CD98 antibodies (scCD98) bound to polymer using click chemistry	CD98 siRNA	Inflammatory bowel disease (IBD)	Targeted NPs reduce levels of CD98 in cell culture and in IBD mice models. Further oral administration of targeted NPs decreases the severity of colitis in mice	[139]
**Lipid-based Particle**	DSPC, DMG-PEG, DSPE-PEG-Ome, cationic lipid 10 (EA-PIP), surface modification using the ASSET platform	Anti-epidermal growth factor receptor (EGFR) antibodies	HPV E6/E7 oncoprotein (E6/E7) siRNA	Human papillomavirus (HPV)-induced head and neck cancer	Targeted NPs induced more apoptosis in cancer cells in vitro compared to untargeted NPs. Further treatment with targeted NPs reduced the tumor size by 50% compared to the control group in a xenograft HPV- positive tumor model	[140]
	DOTAP, cholesterol, protamine, calf thymus DNA, antibody bound via DSPE-PEG-Mal linker molecule	Anti-Th17 antibodies	Tetraspanin 1 (TSPAN1) siRNA	Gastric cancer prevention	Treatment with targeted NPs enabled a longer tumor free time in a gastric cancer mouse model	[144]
	9-arginine peptide, O,O’-dimyristyl-N-lysyl glutamate, cholesterol, DSPE-PEG2000, Rho-DOPE, antibody bound via DSPE-PEG2000-MAL linker molecule	Cetuximab, chimeric antibody against EGFR	Anti-tumorBcl-2/survivin siRNA	Metastasized tumors, especially in the lungs	Efficient siRNA delivery to metastasized tumors, especially in the lungs, resulting in slower tumor growth and extended lifespan in the cancer mouse group treated with targeting NPs	[147]
	Dlin-MC3-DMA, DSPC, cholesterol, DMG-PEG200, DSPE-PEG-Ome and DSPE-PEG200, surface modification using the ASSET platform	Anti-Ly6C antibodies	Interferon regulatory factor 8 (IRF8) siRNA	Inflammatory bowel disease (IBD)	Good therapeutic effects through the intravenous administration of targeted NPs in the treatment of IBD in mice	[148]

## Data Availability

Not applicable.

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
