# Peer review of "Surface Design Options in Polymer- and Lipid-Based siRNA Nanoparticles Using Antibodies"

_ijms, 2022, doi:10.3390/ijms232213929_

Round 1

Reviewer 1 Report

Grammar is inconsistent throughout, appearing better in some sections. Appears to be a useful review of pertinent background, followed by several examples including relevant clinical results. Would benefit not only from review of English grammar and style, but editing to sharpen topics, structure, and flow of paper.

Author Response

International Journal of Molecular Sciences (ISSN 1422-0067)  

Manuscript ID ijms-1857442  Review Type

Title:

Surface design options in polymer- and lipid-based siRNA  nanoparticles using antibodies

The authors would like to thank the editors and reviewers for their time and their thoughtful and thorough comments on our submitted review article. We highly appreciate your remarks and adjusted our paper accordingly. Our changes and refinements are marked up using the “Track Changes” function in our revised article.

Reviewer #1

  • Reviewers comment: Grammar is inconsistent throughout, appearing better in some sections. Appears to be a useful review of pertinent background, followed by several examples including relevant clinical results. Would benefit not only from review of English grammar and style, but editing to sharpen topics, structure, and flow of paper.

  • Authors answer: Thanks a lot for your valuable evaluation of our review article. We addressed your comments and think it improved our manuscript considerably. We consulted an English lector to revise and correct our language and hope our adjustments suit you. Furthermore, we have revised the structure of the review, sharpened the chapters and also better organized the flow of the review, as can be seen, for example, on page 2 in lines 78-83. Further improvements in the structure can be found throughout the review through the track change function.

Reviewer 2 Report

Only 3 comments:

1. The title is somewhat misleading. It should be changed to explicitly mention antibodies e.g, .... surface-modification by antibodies. 

2. The obstacles in the synthesis and stability of surface-modified antibodies, which have high MW and 2d/3d configuration, should be discussed.

3. The major difference between the fast/immediate release of the cargo by liposomal delivery systems vs. polymeric NPs should be highlighted.

Author Response

International Journal of Molecular Sciences (ISSN 1422-0067)  

Manuscript ID ijms-1857442  Review Type

Title:

Surface design options in polymer- and lipid-based siRNA  nanoparticles using antibodies

The authors would like to thank the editors and reviewers for their time and their thoughtful and thorough comments on our submitted review article. We highly appreciate your remarks and adjusted our paper accordingly. Our changes and refinements are marked up using the “Track Changes” function in our revised article.

Reviewer #2

  • Reviewers comment: Only 3 comments:
  1. The title is somewhat misleading. It should be changed to explicitly mention antibodies e.g, .... surface-modification by antibodies. 
  2. The obstacles in the synthesis and stability of surface-modified antibodies, which have high MW and 2d/3d configuration, should be discussed.
  3. The major difference between the fast/immediate release of the cargo by liposomal delivery systems vs. polymeric NPs should be highlighted.

  • Reviewers comment: The title is somewhat misleading. It should be changed to explicitly mention antibodies e.g, .... surface-modification by antibodies. 
  • Authors answer: Thanks for the helpful advice on improving the title. To make the content of the review more obvious in the title we have therefore named the title: Surface design options in polymer- and lipid-based siRNA nanoparticles using antibodies

  • Reviewers comment: The obstacles in the synthesis and stability of surface-modified antibodies, which have high MW and 2d/3d configuration, should be discussed.
  • Authors answer: To address this comment, we have added a new chapter to the review titled "Nanoparticle functionalization with antibodies", which can be found on page10 starting from line 443. Here we will go into more detail about the general structure of antibodies. We discuss the importance of the orientation of the antibodies on the surface of the nanoparticles for the recognition of antigens and highlight the differences in the binding possibilities. Furthermore, we refer to the review by [129] Marques et al. 2020; 320:p180-200; J Control Release in which the functionalization of nanoparticles by antibodies is discussed in more detail and several surface modification strategies using antibodies are presented.

  • Reviewers comment: The major difference between the fast/immediate release of the cargo by liposomal delivery systems vs. polymeric NPs should be highlighted.

  • Authors answer: We have highlighted the difference in release based on the route of application between polymer and lipid-based nanoparticles on page 8 in lines 310 to 331. In order not to exceed the scope of the review, we have kept the differences very general.

Reviewer 3 Report

This is an review article about nanoparticle-based siRNA delivery system. There are a lot of review articles (about 50 papers in 2022) about the same topics. The novelty of this review should be shown. I think both surface modification and lipid/polymer components are important. And, the demonstration of the in vivo therapeutic effects is also indispensable. Table 1 should include these information. Because this paper contains many grammatical mistakes, the authors should check the whole manuscript.

Author Response

International Journal of Molecular Sciences (ISSN 1422-0067)  

Manuscript ID ijms-1857442  Review Type

Title:

Surface design options in polymer- and lipid-based siRNA  nanoparticles using antibodies

The authors would like to thank the editors and reviewers for their time and their thoughtful and thorough comments on our submitted review article. We highly appreciate your remarks and adjusted our paper accordingly. Our changes and refinements are marked up using the “Track Changes” function in our revised article.

Reviewer #3

  • Reviewers comment: This is an review article about nanoparticle-based siRNA delivery system. There are a lot of review articles (about 50 papers in 2022) about the same topics. The novelty of this review should be shown. I think both surface modification and lipid/polymer components are important. And, the demonstration of the in vivo therapeutic effects is also indispensable. Table 1 should include these information. Because this paper contains many grammatical mistakes, the authors should check the whole manuscript.
  • Authors answer: Thanks a lot for your valuable evaluation of our review article. We addressed your comments and think it improved our manuscript considerably. To address the comment, we expanded the table 1 and added much more desired information, such as the main components of the nanoparticle matrix or the therapeutic effects of the nanoparticle administration. The improved table 1 now provides a much better overview than before and shows the novelty of this review in an appropriate manner. Additionally, we consulted an English lector to revise and correct our language and hope our adjustments suit you.

Round 2

Reviewer 1 Report

Much improved, and will make a contribution to the field. 

Reviewer 3 Report

The authors replied to my comments properly.